# Urban Sustainability at Risk Due to Soil Pollution by Heavy Metals—Case Study: Volos, Greece

**Panagiotis-Stavros C. Aslanidis [1],***  **and Evangelia E. Golia [2,3]**

1    Department of Planning and Regional Development, University of Thessaly, Pedion Areos, 383 34 Volos, Greece
2    Laboratory of Soil Science, Faculty of Agriculture, Forestry and Natural Environment, School of Agriculture, Aristotle University of Thessaloniki, University Campus, 541 24 Thessaloniki, Greece; egolia@auth.gr
3    Laboratory of Soil Science, Department of Agriculture, Crop Production and Rural Environment, University of Thessaly, Fytokou Street, 384 46 Volos, Greece
*    Correspondence: panagiotis.aslanidis@yahoo.gr

**Abstract:** The focus of this case study is the meticulous observation of urban soil pollution by heavy metals (HMs), or, alternatively, potentially toxic elements (PTEs). The study took place in the urban center of Volos, Greece. Moreover, 248 soil samples were collected during 2018–2021 (62 samples annually), while 3.65 km$^2$ was, approximately, the study area. The breakdown of total concentrations took place for the interpretation of different soil parameters, also according to mean values and medians of the total concentrations of HMs, the following decreasing order was monitored: Mn > Zn > Cr > Ni > Cu > Pb > Co > Cd. During the 4-year study, an increasing trend of metal concentration was observed (for each year compared to the previous one). Furthermore, the imaginary triangle, which was observed, is bordered by the historic train station, the two city bus and intercity coach stations and the commercial harbor. Statistical analysis was implemented in order to interpret the exceedances of HMs concerning the Directive 86/278/EEC. Principal component analysis (PCA) is an additional technique that was conducted because of the correlations and interdependences between the HMs. A strong correlation was observed between the HMs, but mainly between Cd and Zn, which is probably due to their common origin. During the COVID-19 pandemic, significant changes in metal concentrations were observed in different parts of the city, due to the limited movement of motorized wheeled vehicles, but also due to the long operating hours of the heating systems in the residential area. Further research is needed in the future in order to identify the sources of pollution and to find possible ways to reduce it. All in all, urban soil pollution by HMs is a great conundrum of the environmental aspect of sustainability.

**Keywords:** potentially toxic elements; soil contamination; principal component analysis; COVID-19



## 1. Introduction

Various driving forces set the environmental equilibrium into instability, for example, population growth, urbanization, (de-)industrialization, energy poverty, and COVID–19, to name but few. In 1968, the "World Problematique" was stated by the Club of Rome, an institution that called for action to address different enigmas of the current state of society, environment, and economy. The publication "The limits to Growth" addressed the necessity of taking measures in order to preserve soil quality and fertility, of course, by not abusing them due to the capitalization of agriculture [1]. Hence, one proper quote to characterize the present situation is "a world of abundance, not one of limits, pollution, and waste" [2].

Heavy metals (HMs) or potentially toxic elements (PTEs) have detrimental effects to people. The present study will shed light on eight HMs: copper (Cu), zinc (Zn), cadmium (Cd), manganese (Mn), lead (Pb), nickel (Ni), chromium (Cr), and cobalt (Co). The elements Cu, Zn, Mn, Ni and Co in small quantities (traces) are necessary for the proper functioning and growth of living organisms and that is why they are called trace elements. However,

when they exceed a critical value, they may pose risks to human health and the environment. They provoke adverse effects such as diseases and illnesses [3–10].

The transition from Fordism to post-Fordism had, as a result, an alteration of the economic system [11–13]. There are several socio–economic parameters that have been destabilized, such as population proliferation and urban sprawl. The impact of urbanization and industrialization leads to high levels of HMs in urban and agricultural environments [14–19]. Pollution sources could be places with intense concentrations of commercial and entrepreneurial activity. For example, a maritime port, a railway station, industrial areas, and bus terminals are fields of operations with possible overaccumulation of HMs [18,20,21].

Furthermore, the present health crisis of COVID-19 and the two lockdowns during 2020–2021 had, as an outcome, the alteration of human normality [22,23]. The differentiated normality of human lives could throw into question periods of the heating of houses; in parallel, one ought to take into consideration the phenomenon of energy poverty. Energy poverty is looming, and households might use environmentally unfriendly ways of heating, such as rudimentary and unsophisticated wood fuel type, as an alternative heating source [24,25]. On the other hand, during this period, the movement of wheeled vehicles on main roads was significantly reduced, but also at the entrances and exits of cities, as the movement of citizens was prohibited [26,27]. As a result, pollutants such as HMs stick to greenhouse gases (GHGs) and might exacerbate climate change, but also, when GHGs sink to soil, it is possible that this circle of HMs might aggravate soil's quality as well, and put the environmental stability at risk.

Since the birth of industrialization, soil quality has fallen prey to the mechanized production system, because of the rampant use of pesticides and chemical fertilizers [28]. Hence, the necessity of circularity was unambiguous and unavoidable. Concurrently, a circular economy is not a panacea for sustainability, but more actions should be taken to ultimately achieve a sustainable future [29,30].

Furthermore, the World Commission on Environment and Development (WCED), in 1987, focused specifically on two important factors: the 'Urban Challenge' and 'Call for Action' [28]. It can be stated that WCED questioned urban sustainability; however, it was not the first time, since these core values were introduced.

The disruption of urban soils by anthropogenic activity is apparent in urban agglomerations and their outgrowths [16]. The sealing of urban soils can be found in former commercial or industrial sites, also known as brownfields [31–35]. Lately, the intensity of human commercial activities aggravated urban soil conditions via—organic or inorganic—pollutants [36]. Regarding the present study, light is shed exclusively on inorganic pollutants: heavy metals. Biodiversity and vulnerable social groups seem to have powerful interplay; while, at the same time, socio-economic forces—travel, trade, and transportation—push this connection towards instability [37,38]. The above socio-economic pressures are critical to the stability of the preservation of ecosystem services.

Urban soil or even air pollution by HMs has been a significant focus in different areas in Greece, for instance in former mines, near carriage roads, and old factories [39–43]. Furthermore, the most populous coastal cities in Greece (Athens, Thessaloniki, Patras, and Volos) share common local characteristics, such as the presence of maritime ports, which contribute to urban pollution by HMs [18,35,44–50].

The objectives of the present study were: (a) the recording of the levels of HMs in the city of Volos during the 4-year study, (b) the revelation of interrelationships between metal elements, using statistical tools, (c) the monitoring of COVID-19 impacts on HMs concentrations, and (d) the investigation of possible sources of pollution in the study area. The originality of the present study is based on the categorization of driving forces that put the studied area into environmental instability.

## 2. Materials and Methods

### 2.1. Site Description

The city of Volos is circumscribed by the mythical mountain Pelion, while its exodus to Aegean Sea is through Pagasiticos Gulf. The micro-climate conditions that exist in Volos could be described as peculiar and particular, at the same time. [18,50]. Furthermore, there are several industrial areas in the vicinity of the city. For instance, the cement factory in Agria, and a steel plant in the industrial area [18]. Additionally, two parameters that have special relevance to the present study are the maritime port and the historic train station, which are the focal points of Volos' citizens.

### 2.2. Soil Sampling

The soil sampling Scheme 1 was achieved via the collection of 248 soil samples in the time span 2018–2021, meaning that there was a selection of 62 soil samples—with 3 surface sub-samples of 0–20 cm depth from the same sampling spots—per annum [18]. The sampling took place during four consecutive years, from 62 points shown in the map below.

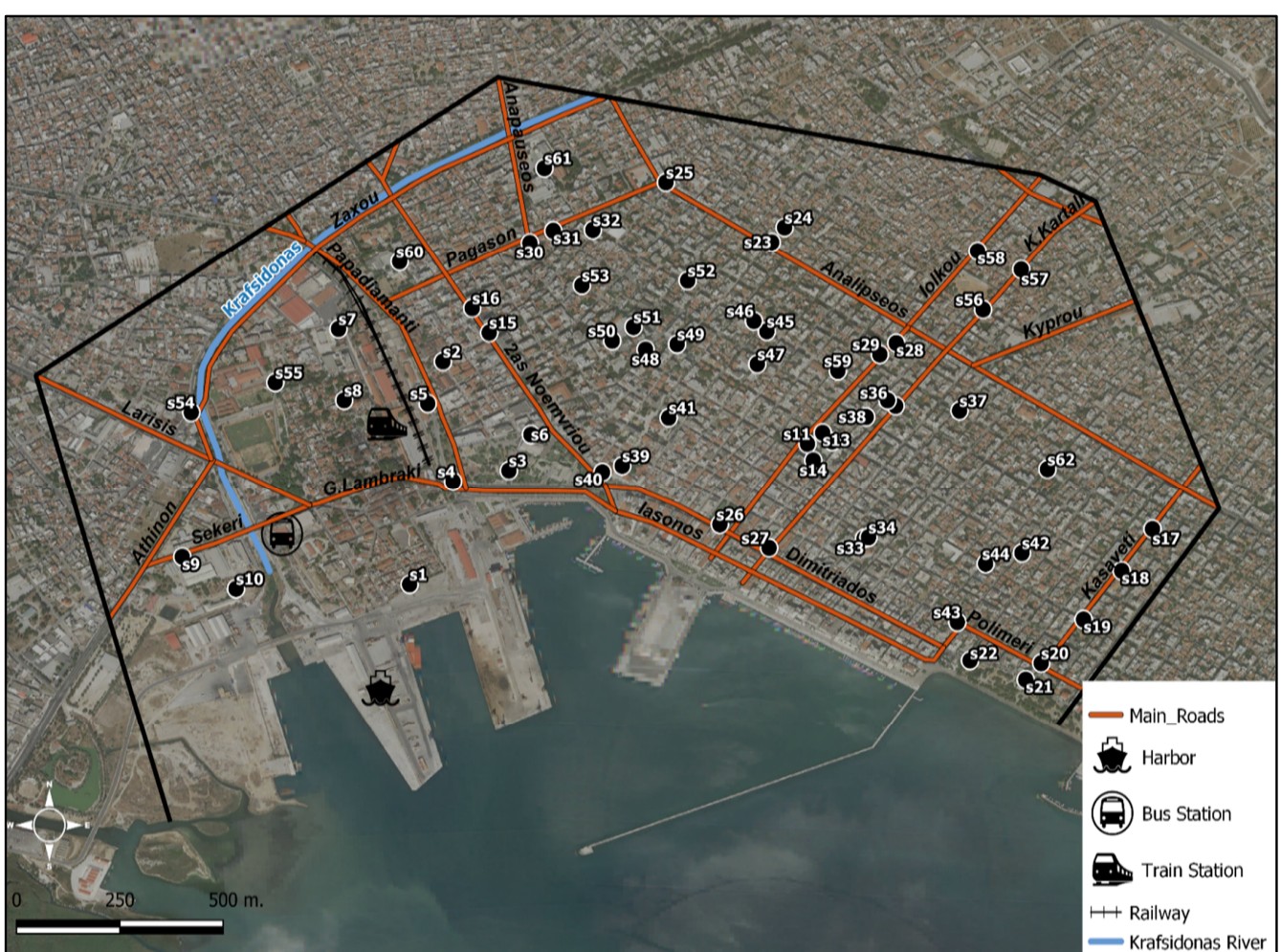

**Scheme 1.** Sampling sites in the commercial and residential city of Volos.

Soil samples were compiled by various green places, such as: pocket-sized parks, playgrounds, and parterres. In addition, special focus was given to adjacent places to the port, the train station and the two bus stations, all three of which constitutes an imaginary triangle, at the core of which the standards of urban soil pollution were monitored. Samples were categorized into sub-groups as: imaginary triangle which is bordered by the historic train station, the two city bus and intercity coach stations and the commercial harbor

(s1–s10); residential areas (s39–s53); and boulevards and main roads as the rest of sampling points (s11–s38 and s54–s62).

The sampling was completed via wooden sieves, the samples were put into plastic bags and then they were transported to the soil science lab, where the proper procedure was performed. The procedure has the following steps, firstly there was the air-drying of the samples via sieve of 2 mm, then physicochemical analyses were carried out [51–53].

### 2.3. Chemical Analysis of Soil Samples

Next, the method of Aqua Regia was the backbone of the analysis of several parameters: pH values, electrical conductivity, organic matter content, cation exchange capacity (CEC), pseudo-total content, and the percentages of sand, silt, and clay [51–53]. Extractive methods [54] with strong acidic and alkaline solutions were used to quantify the total concentrations of heavy metals. The Aqua Regia method was used to determine the pseudo-total concentration of metals, along with the pollution indices values [18]. Pseudo-total and total concentrations did not show statistically significant differences ($p > 0.05$) in any metallic element throughout the study. HMs were analyzed by an atomic absorption spectrophotometer using flame and/or graphite furnace equipment. The detection limits were 0.1, 0.09, 0.09, 0.1, 0.08, 0.2, 0.1, and 0.15 (mg $L^{-1}$) for Cu, Zn, Cd, Mn, Pb, Ni, Cr, and Co, respectively, according to ISO 11446 [55]. At the same time, there is a mention of the soil quality guidelines for the protection of environmental and human health of the Canadian Council of Ministers of the Environment [56].

### 2.4. Statistical Analysis

Foremost, the outputs of four tests were checked via SPSS 26.0: Levene's test of homogeneity, ANOVA, Kolmogorov–Smirnov test, and Kruskal–Wallis test; these outputs can be found in the supplementary material with great detail, while their graphical depiction is illustrated in Appendices A and B. The first test is Levene's, which has as null hypothesis that there is homogeneity of variance. Secondly, ANOVA shows if the means are equal. Appendix A illustrates the output of Kolmogorov–Smirnov and Appendix B depicts the boxplots as a graphical glimpse of Kruskal–Wallis test, because the line in box-plots mean the median of HM total concentrations. The last two tests are Kolmogorov–Smirnov and Kruskal–Wallis, the former shows if the data follow normal distribution, while the latter's null hypothesis implies that the data follow the same distribution, or, alternatively, Kruskal–Wallis test reveals that the medians are equal. All the aforementioned tests were checked at 5% level of significance (a = 0.05%). The results showed that that the data did not follow normal distribution (K–S test), while Cu, Mn, Pb, Ni have same distribution (K–W test). These four tests were necessary for the other three tests that were investigated: Tukey's honestly significant difference (Tukey's HSD), Fisher's least significance difference (Fisher's LSD), and Games–Howell, in order to check the pairs of samples, the results of which can be found in the supplementary material (Tables S1–S8).

In addition, the percentage changes are highlighted in Figure 1. Furthermore, Pearson's correlation analysis was necessary to examine the relationships among the HMs [57], but a second important reason for correlation analysis is its importance in cluster analysis (CA) and principal components analysis (PCA). CA and PCA were executed aiming to identify the connections between HMs and to examine potential sources of origin. CA in depicted in Figure 2a which is a dendrogram that is the output of Ward's method; in addition, Figure 2b is the scree plot from PCA. While PCA (Figure 3a,b) was executed via the command in SPSS 'Varimax Rotation', because Varimax rotation enables a more proper explanation [47]. The main reason of the importance of executing PCA is because one could examine fewer components. For example, in the present study, instead of 8 HMs via PCA one could interpret only 2 components. Meaning that only two sets can clarify the potential sources of HMs, not at 100%, but around 85% (in the present study) of the initial variance. These two sets (components) have eigenvalues with values over 1.00, which is used as a rule of thumb.

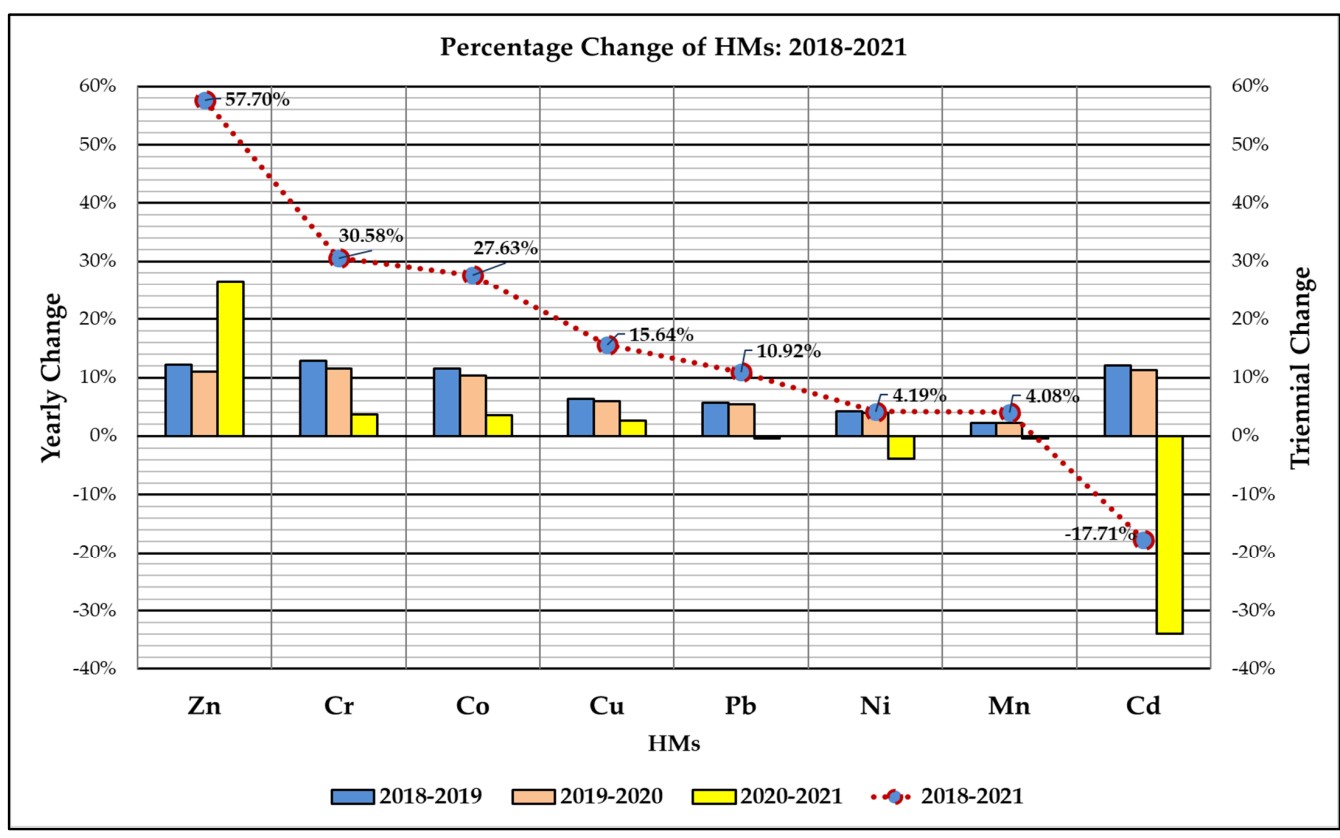

**Figure 1.** Yearly and triennial percentage change in the total concentrations of HMs from 2018 to 2021.

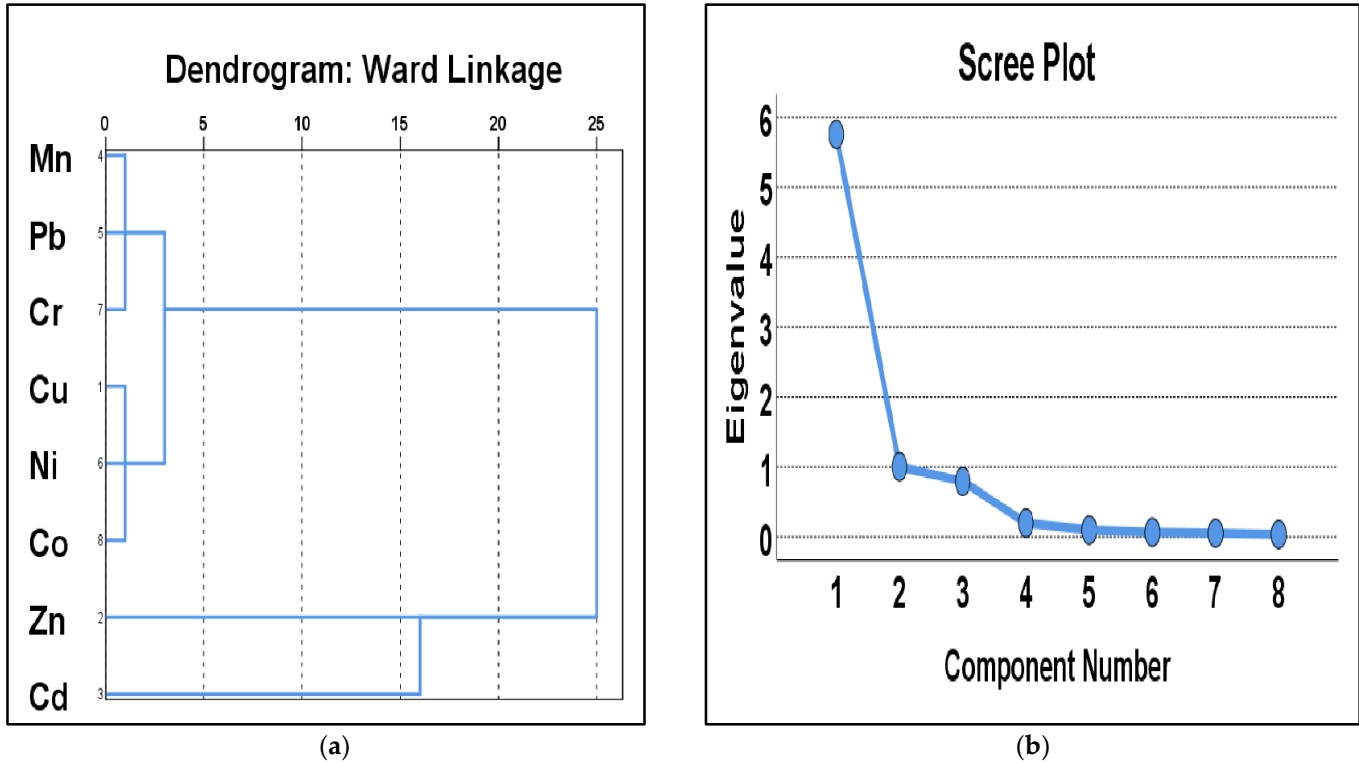

**Figure 2.** (**a**) Output of cluster analysis via dendrogram with Ward's linkage; (**b**) output of principal component analysis via Scree plot, which shows the selected components (groups) that have values over '1'.

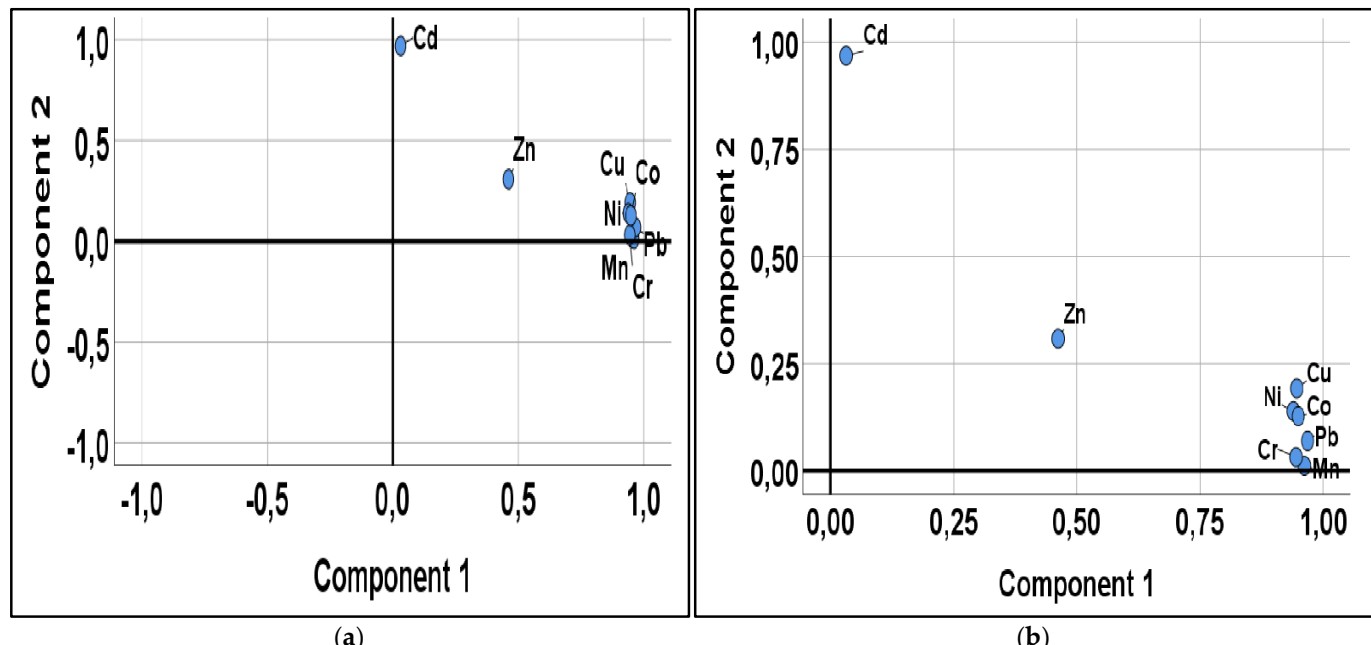

(**a**)  (**b**)

**Figure 3.** (**a**) Output of principal components analysis with Varimax rotation; (**b**) zoom in on the results of principal components analysis.

## 3. Results

### 3.1. Physicochemical Properties of Soil Samples

It can be stated that the majority of soils in Table 1 have a sandy texture of 56.7% according to the mean value. However, silt reached a level of 24.1% of soil content, while the rest of the soils have high clay content, of approximately 19.2%. About 20% of the soil samples showed high clay values, while the largest percentage (almost 60%) had a sandy texture. Based on soil texture, the soil samples ranged from sandy loam to clayey.

**Table 1.** Descriptive statistics of physicochemical parameters of the soil samples (four-year study, n = 248).

| Physicochemical Parameters | | pH | OM | EC | CaCO$_3$ | Sand | Silt | Clay |
|---|---|---|---|---|---|---|---|---|
| | | (1:1) | (%) | (μS cm$^{-1}$) | (%) | (%) | (%) | (%) |
| Mean | | 7.44 | 2.44 | 3235.58 | 14.57 | 56.70 | 24.14 | 19.16 |
| Std. Error of Mean | | 0.06 | 0.12 | 176.87 | 0.33 | 1.73 | 0.58 | 1.65 |
| Median | | 7.40 | 2.50 | 2700.50 | 14.25 | 60.50 | 24.00 | 14.00 |
| Mode | | 7.00 | 3.00 | 1998.00 | 13.40 | 67.00 | 24.00 | 14.00 |
| Std. Deviation | | 0.44 | 0.91 | 1392.69 | 2.56 | 13.62 | 4.58 | 13.00 |
| Variance | | 0.19 | 0.82 | 1,939,584.78 | 6.55 | 185.39 | 21.02 | 169.04 |
| Skewness | | 0.71 | −0.13 | 0.75 | 0.23 | −0.93 | 0.36 | 1.10 |
| Std. Error of Skewness | | 0.30 | 0.30 | 0.30 | 0.30 | 0.30 | 0.30 | 0.30 |
| Kurtosis | | 1.42 | −0.68 | −0.50 | −0.68 | −0.07 | 0.71 | −0.01 |
| Std. Error of Kurtosis | | 0.60 | 0.60 | 0.60 | 0.60 | 0.60 | 0.60 | 0.60 |
| Range | | 2.35 | 4.10 | 5834.00 | 10.90 | 56.00 | 24.00 | 50.00 |
| Minimum | | 6.57 | 0.30 | 1123.00 | 9.60 | 22.00 | 12.00 | 2.00 |
| Maximum | | 8.92 | 4.40 | 6957.00 | 20.50 | 78.00 | 36.00 | 52.00 |
| Sum | | 461.20 | 151.50 | 200,606.10 | 903.50 | 3515.50 | 1496.50 | 1188.00 |
| Percentiles | 25 | 7.13 | 1.70 | 2027.00 | 12.50 | 49.50 | 21.00 | 12.00 |
| | 75 | 7.73 | 3.10 | 4387.75 | 16.73 | 67.00 | 26.00 | 27.00 |

Another crucial parameter of soil content is organic matter (OM). OM has a range of 4.1%, with 0.3% and 4.4% as minimum and maximum values, respectively. In addition, soil electrical conductivity (EC) could reach values between 1123 and 6957 μS cm$^{-1}$, with mean 3235.58 μS cm$^{-1}$ and median 2700.50 μS cm$^{-1}$, respectively. Soil samples are, therefore, mainly sandy and barren, as their organic matter content is low. The physicochemical properties of soil samples have the typical characteristic values of most urban soils [18,35,44–50]. Soil samples located near the sea, i.e., in the southern part of the city, have higher values of electrical conductivity, due to the possible presence of salts [18]

### 3.2. Levels of Potentially Toxic Elements

The statistical characteristics of the HMs of the present four–year study in Volos are illustrated in Table 2. The mean and median values of total concentrations of HMs in soils from residential and commercial neighborhoods can be displayed in a decreasing order as: Mn > Zn > Cr > Ni > Cu > Pb > Co > Cd. The limits of Directive 86/278/EEC and the Canadian Council of Ministers of the Environment are displayed for further interpretation of HMs' concentrations [56,58].

**Table 2.** Descriptive statistics of total concentrations of the eight HMs and comparison with Directive 86/278/EEC for agricultural soils and soil quality guidelines for residential areas from CCME [56,58].

| Heavy Metals | | Cu | Zn | Cd | Mn | Pb | Ni | Cr | Co |
|---|---|---|---|---|---|---|---|---|---|
| | | mg·kg$^{-1}$ | | | | | | | |
| Minimum | | 20.13 | 67.80 | 0.12 | 248.80 | 4.86 | 23.19 | 15.54 | 2.78 |
| Mean | | 55.08 | 144.32 | 0.88 | 666.78 | 36.39 | 67.28 | 96.89 | 23.37 |
| Median | | 54.67 | 134.25 | 0.90 | 701.00 | 40.90 | 64.24 | 106.99 | 22.47 |
| Mode | | 59.00 | 121.00 | 1.10 | 899.00 | 44.80 | 56.13 | 133.74 | 36.19 |
| Maximum | | 92.30 | 304.00 | 1.80 | 1129.00 | 60.29 | 122.11 | 178.20 | 42.80 |
| Std. Deviation | | 18.11 | 39.68 | 0.28 | 246.62 | 16.68 | 26.86 | 33.38 | 10.14 |
| Variance | | 327.97 | 1574.13 | 0.08 | 60,821.37 | 278.17 | 721.49 | 1113.96 | 102.75 |
| Skewness | | 0.35 | 0.93 | 0.00 | −0.28 | −0.42 | 0.37 | −0.37 | −0.02 |
| Kurtosis | | −1.01 | 1.01 | 0.04 | −1.31 | −1.19 | −0.72 | −0.88 | −0.96 |
| Range | | 72.17 | 236.20 | 1.68 | 880.20 | 55.43 | 98.92 | 162.66 | 40.02 |
| Percentiles | 25 | 39.10 | 116.00 | 0.70 | 424.10 | 17.66 | 44.91 | 67.14 | 15.62 |
| | 75 | 68.79 | 167.00 | 1.09 | 897.00 | 51.64 | 85.40 | 122.45 | 32.47 |
| Directive 86/278/EEC [58] | | 50–140 | 150–300 | 1.0–3.0 | - | 50–300 | 30–75 | - | - |
| Soil Quality Guidelines [56] | | 63.00 | 250.00 | 10.00 | - | 140.00 | 45.00 | 64.00 | 50.00 |

Additionally, there is a common pattern in Table 3 for the HMs: they have lower values than the maximum permitted vales, either for their mean, or for their median values of HMs total concentrations. Moreover, the acceptable limits of Cd, Cu, Co, Pb, and Zn were checked based on the European Council Directive 86/278/EEC [58]. The mean concentrations of Zn and Pb are 144 and 36 mg·kg$^{-1}$, respectively, while the limit of the European directive is 300 mg·kg$^{-1}$ for each of them. Additionally, the means of Cu and Ni are 55 and 67 mg·kg$^{-1}$ respectively, while the European limit is 140 mg·kg$^{-1}$ and 75, accordingly. Finally, there is Cd's mean value, which is 0.88, while the limit is 3 mg·kg$^{-1}$. However, Ni and Zn have some values that exceed these standards. The former has plenty of values each year that surpass the maximum permitted level: 95 values out of 248 during 2018–2021, while the latter has only one value across the four years, and it can be found in 2021. This environmental destabilization is also shown in Appendix B, in which there is a rising trend in HM concentrations due to the existence of outliers (extreme values), especially in 2020 and 2021 for these two specific HMs during COVID-19 era.

**Table 3.** Extreme values of the HMs according to Directive 86/278/EEC [58].

| Extreme Values | | | Zn | Cd | Pb | Ni | Cu |
|---|---|---|---|---|---|---|---|
| | | | mg·kg$^{-1}$ | | | | |
| | Mean | | 144.32 | 0.88 | 36.39 | 67.28 | 55.08 |
| | Median | | 134.25 | 0.90 | 40.90 | 64.24 | 54.67 |
| | Directive 86/278/EEC | | 150–300 | 1.0–3.0 | 50–300 | 30–75 | 50–140 |
| | | | No. of samples out of 62 (samples per year) | | | | |
| Extreme Values | 2018 | Lower | 54 | 44 | 49 | 6 | 31 |
| | | Higher | 0 | 0 | 0 | 23 | 0 |
| | 2019 | Lower | 50 | 34 | 45 | 5 | 31 |
| | | Higher | 0 | 0 | 0 | 24 | 0 |
| | 2020 | Lower | 40 | 23 | 44 | 4 | 28 |
| | | Higher | 0 | 0 | 0 | 24 | 0 |
| | 2021 | Lower | 10 | 54 | 43 | 5 | 27 |
| | | Higher | 1 | 0 | 0 | 24 | 0 |

Table 4 illustrates the mean values of HMs each year, which seem to have a rising trend for most of them, for example, Cu, Zn, Mn, Pb, Cr, and Co. On the other hand, while Cd and Ni have an increasing pattern from 2018 to 2020, there is an abrupt decline in 2021, especially for Cd concentrations.

**Table 4.** Means values of total concentrations of HMs from 2018–2021.

| Means of HMs | Cu | Zn | Cd | Mn | Pb | Ni | Cr | Co |
|---|---|---|---|---|---|---|---|---|
| | mg·kg$^{-1}$ | | | | | | | |
| 2018 | 50.69 | 116.72 | 0.84 | 649.22 | 34.01 | 64.57 | 82.57 | 20.22 |
| 2019 | 53.91 | 131.02 | 0.94 | 663.71 | 35.94 | 67.28 | 93.24 | 22.56 |
| 2020 | 57.12 | 145.47 | 1.05 | 678.49 | 37.87 | 69.99 | 103.95 | 24.90 |
| 2021 | 58.62 | 184.06 | 0.69 | 675.72 | 37.73 | 67.28 | 107.82 | 25.80 |

Moreover, the depiction of yearly and triennial percentage changes is depicted in Figure 1. For Zn, Cr, Co, and Cu, there are only positive yearly percentage changes, while Pb, Ni, Mn, and Cd seem to have positive yearly percentage changes from 2018 to 2020, but in 2021 a decline appears among them. Furthermore, referring to the triennial percentage change, the greatest changes happened in Zn and Cd. To focalize, Zn skyrocketed by about 58% during 2018–2021 at the imaginary triangle (s1–s10); on the contrary, Cd plummeted by 18% at the imaginary triangle and in residential samples (s39–s53).

*3.3. Statistical Analysis*

Firstly, in Table 5 there are the outputs of: Levene's test of homogeneity, ANOVA, the Kolmogorov–Smirnov test, and the Kruskal–Wallis test. The Levene test's null hypothesis stipulates that there is homogeneity of variance. The null hypothesis of ANOVA indicates that the means are equal. Furthermore, the Kolmogorov–Smirnov test shows if the data follow normal distribution, while the null hypothesis of the Kruskal–Wallis test indicates whether the data follow the same distribution, or, alternatively, the Kruskal–Wallis test reveals that the medians are equal.

From the eight HMs of our study, only Cd rejected the null hypothesis of Levene's test. The meaning of this statement is that Cd shows heterogeneity of variance. Levene's test is essential for two reasons. One reason why is for the use of post hoc tests, such as Tukey and Fisher's LSD, or Games–Howell. The second reason is that if there is no homogeneity of variance, the data do not follow the normal distribution; thus, Kruskal–Wallis should be applied.

**Table 5.** Output of tests: Levene, ANOVA, Kolmogorov–Smirnov, and Kruskal–Wallis. The $p$-values are shown.

| Criteria | | Cu | Zn | Cd | Mn | Pb | Ni | Cr | Co |
|---|---|---|---|---|---|---|---|---|---|
| Test of homogeneity of variances | Based on Mean | 1.000 | 0.102 | 0.004 | 0.644 | 0.888 | 0.398 | 0.580 | 0.175 |
| | Based on Median | 1.000 | 0.111 | 0.015 | 0.694 | 0.943 | 0.495 | 0.770 | 0.243 |
| | Based on Median and with adjusted df | 1.000 | 0.111 | 0.016 | 0.694 | 0.943 | 0.495 | 0.770 | 0.243 |
| | Based on trimmed mean | 1.000 | 0.101 | 0.009 | 0.639 | 0.900 | 0.414 | 0.556 | 0.166 |
| ANOVA | | 0.069 | 0.000 | 0.000 | 0.910 | 0.536 | 0.740 | 0.000 | 0.009 |
| Kolmogorov–Smirnov | | 0.000 | 0.000 | 0.011 | 0.000 | 0.000 | 0.002 | 0.000 | 0.003 |
| Kruskal–Wallis | | 0.025 | 0.000 | 0.000 | 0.743 | 0.162 | 0.458 | 0.000 | 0.010 |

The results derived from ANOVA illustrate an interesting pattern. Four HMs (Zn, Cd, Cr, and Co) have rejected the null hypothesis of same means and there is statistically significant difference between the means of each year in every HM, at a level of significance of 5% ($p < 5\%$). On the other hand, the other four HMs (Cu, Mn, Pb, and Ni) did not reject the null hypothesis ($p > 5\%$).

The Kolmogorov–Smirnov test shows that none of the HMs follow normal distribution at a level of significance of 5%. One reason why is because soil samples were collected in different areas of Volos, while if they had been collected in the same neighborhood, they would probably depict normal distribution.

Seven of the HMs showed homogeneity of variance, as has already been proven. This is the reason why two preferable tests (Tukey and Fisher's LSD) are applicable in order to extract the pairs that have significant difference. On the other hand, for the reason why Cd has heterogeneity of variance, a test that assumes the aforementioned hypothesis is more appropriate, such as the Games–Howell test.

The results of the Tukey, Fisher's LSD, and Games–Howell tests are illustrated in the supplementary material. Mn, Pb, and Ni have no statistically significant difference, as was also implied by the output of ANOVA. There are significant differences between pairs in Cu, Co, and Cr, either for Tukey's test or LSD. Tukey and LSD tests have statistically significant differences in Zn samples, except the years 2018 and 2019 in Tukey's test. Cd was the only HM that was checked via the Games–Howell test, because test Levene showed heterogeneity of variance. Hence, according to Games–Howell test, there is statistically significant difference for each pair.

3.3.1. Correlation Analysis

The correlation coefficients of the eight HMs and soil properties and characteristics are depicted in Table 6. In general, there are strong correlations ($r > 0.8$) and statistically significant differences ($p < 0.5$ and $p < 0.01$) between HMs. However, Zn and Cd show a slight correlation ($0 < r < 0.4$) with the other HMs, but that is still statistically significant. It should also be stated that there is no statistically significant difference between Cr and Cd, while there is low correlation and statistically significant difference ($r = 0.15$ and $p < 5\%$) between Co and Cd. There is no statistically significant difference between pH and the other properties; this phenomenon indicates that soil has no effect on the content of HMs; a similar effect was observed in a similar study [59].

EC has slight negative correlations ($-0.5 < r < -0.3$, $p < 0.01$) with HMs. EC also shows positive correlation ($r = 0.2$) with sandy content, but negative correlation ($r = -0.2$) with clayed texture; both correlations are statistically significant ($p < 0.05\%$). It could be inferred that $CaCO_3$ and Silt have no correlation with the other elements or soil parameters, while OM has statistically significant positive correlation only with Mn ($r = 0.2$, $p < 0.05$). Additionally, soils with a sandy texture have statistically significant negative correlations ($-0.6 < r < -0.4$, $p < 0.01$) with HMs. Sandy soils have a statistically significant positive correlation with EC ($r = 0.2$, $p < 0.05$), statistically significant negative correlations with

silt and clay soil, with r = −0.2 and r = −0.9 and *p*-values < 0.01 and 0.05, respectively. Sandy soils have no correlation with pH, $CaCO_3$, or OM. Soils that could be described by an excessive content of clay follow quite the opposite pattern to the sandy ones. Clay has statistically significant positive correlations with each HM (0.4 < r < 0.6, *p* < 0.1). Clay has statistically significant negative correlation with EC and sandy soils (r = −0.2 and r = −0.9, and *p* < 0.1 and *p* < 0.5, respectively). It should be cited that clay has no significant correlation with pH, $CaCO_3$, silt, and OM.

**Table 6.** Correlation coefficients between HMs and soil physicochemical parameters.

| | Cu | Zn | Cd | Mn | Pb | Ni | Cr | Co | pH | EC | $CaCO_3$ | Sand | Silt | Clay | OM |
|---|---|---|---|---|---|---|---|---|---|---|---|---|---|---|---|
| **Cu** | 1 | | | | | | | | | | | | | | |
| **Zn** | 0.512 ** | 1 | | | | | | | | | | | | | |
| **Cd** | 0.200 ** | 0.115 | 1 | | | | | | | | | | | | |
| **Mn** | 0.896 ** | 0.332 ** | 0.084 | 1 | | | | | | | | | | | |
| **Pb** | 0.900 ** | 0.381 ** | 0.130 * | 0.956 ** | 1 | | | | | | | | | | |
| **Ni** | 0.934 ** | 0.391 ** | 0.176 ** | 0.892 ** | 0.909 ** | 1 | | | | | | | | | |
| **Cr** | 0.862 ** | 0.426 ** | 0.082 | 0.918 ** | 0.928 ** | 0.832 ** | 1 | | | | | | | | |
| **Co** | 0.913 ** | 0.442 ** | 0.157 * | 0.888 ** | 0.914 ** | 0.917 ** | 0.879 ** | 1 | | | | | | | |
| **pH** | −0.049 | −0.167 | −0.255 * | 0.072 | 0.091 | −0.044 | 0.156 | −0.044 | 1 | | | | | | |
| **EC** | −0.551 ** | −0.344 ** | −0.089 | −0.514 ** | −0.524 ** | −0.500 ** | −0.572 ** | −0.560 ** | −0.036 | 1 | | | | | |
| **$CaCO_3$** | −0.019 | −0.083 | −0.078 | −0.001 | −0.001 | −0.044 | −0.079 | −0.010 | −0.019 | 0.193 | 1 | | | | |
| **Sand** | −0.646 ** | −0.686 ** | −0.361 ** | −0.450 ** | −0.540 ** | −0.667 ** | −0.425 ** | −0.547 ** | 0.070 | 0.255 * | 0.140 | 1 | | | |
| **Silt** | 0.015 | 0.139 | 0.029 | 0.032 | 0.015 | 0.004 | 0.001 | −0.048 | 0.093 | 0.072 | −0.239 | −0.299 * | 1 | | |
| **Clay** | 0.671 ** | 0.669 ** | 0.368 ** | 0.459 ** | 0.560 ** | 0.698 ** | 0.444 ** | 0.589 ** | −0.106 | −0.292 * | −0.062 | −0.942 ** | −0.039 | 1 | |
| **OM** | 0.185 | 0.227 | 0.096 | 0.259 * | 0.185 | 0.160 | 0.218 | 0.124 | −0.023 | −0.127 | −0.079 | −0.103 | −0.039 | 0.122 | 1 |

* Correlation is significant at the 0.05 level (2-tailed). ** Correlation is significant at the 0.01 level (2-tailed).

### 3.3.2. Cluster Analysis

The hierarchical dendrogram derived from cluster analysis is illustrated in Figure 2a. In this figure, it could be assumed that there are two major groups, the first of which is composed of two subgroups. The first major cluster has two subclusters, the first subcluster contains Mn, Pb, and Cr, while the other one incorporates Cu, Ni, and Co. The first subcluster (Mn, Pb, and Cr) shows an intense relation between the HMs, while the second one depicts a similar pattern as well, between the HMs (Cu, Ni, and Co). This occurrence may assume a similar source. The second major cluster encloses Cd and Zn; however, there is no intense relation between the two HMs; this phenomenon may put into question the probable different—anthropogenic or natural—source(s). Hence, cluster analysis suggests at least two (and possibly three) different sources of heavy metals.

### 3.3.3. Principal Component Analysis

The output of KMO and Bartlett's test are displayed in Table 7. The value of the KMO test (0.88) could be described as 'meritorious' [60] and postulates the use of PCA as extremely auspicious and advantageous. Furthermore, the Bartlett's test's null hypothesis is rejected (*p* < 0.05), which could be explained as there being substantial correlation in the data meaning the present analysis could be completed.

**Table 7.** Kaiser–Mayer–Olkin test of sampling adequacy and Bartlett's test of sphericity.

| KMO and Bartlett's Test | | |
|---|---|---|
| Kaiser-Meyer-Olkin Measure of Sampling Adequacy | | 0.887 |
| Bartlett's test of sphericity | Approx. chi square | 2795.748 |
| | df | 28 |
| | Sig. | 0.000 |

It can be also stated that, in order to check eight components (equal with the present study's HMs), it is more preferable that PCA be applied, which aims to make the interpretation simpler and of some importance. In Table 8, total variance is explained; it can be determined that the principal components are two, because of the existence of two eigenvalues that surpass the value '1'.

**Table 8.** Output of principal components analysis, eigenvalues, percentage of variance, and cumulative percentage of variance.

| Principal Components Analysis | | | |
|---|---|---|---|
| **Component** | **1** | **2** | **Communalities** |
| Cu | 0.946 | 0.192 | 0.932 |
| Zn | 0.462 | 0.308 | 0.308 |
| Cd | 0.032 | 0.969 | 0.939 |
| Mn | 0.962 | 0.012 | 0.925 |
| Pb | 0.968 | 0.070 | 0.942 |
| Ni | 0.939 | 0.139 | 0.902 |
| Cr | 0.945 | 0.032 | 0.893 |
| Co | 0.949 | 0.127 | 0.917 |
| Eigenvalue | 5.647 | 1.112 | |
| % of Variance | 70.588 | 13.898 | |
| Cumulative % of Variance | 70.588 | 84.486 | |

The same result could be inferred by the scree plot in Figure 2b. A scree plot is practical as a mean when one desires to extract the possible eigenvalues out of this graph. The eigenvalues are illustrated on the vertical axis, while components are displayed on the horizontal axis. It is a rule of thumb that eigenvalues with a value above '1' should be input into the analysis. To put it simply, if there were more eigenvalues above the value of '1', then more components should have been taken into account. In the present study, the first eigenvalue is 5.647, and the second one is 1.112; these two eigenvalues, which are above the value of '1', ought to be extracted. Moreover, the first component consists of 70.58% of variance, while the second one is made up by 13.89%. In a nutshell, instead of monitoring 100% of the variance for the eight components (HMs), the two extracted components (clusters) could be explained briefly and properly only by taking into consideration 84.4% of the initial variance.

The first component is made up of Cu, Mn, Pb, Ni, Cr, and Co because they have values above 0.9, but Zn and Cd are excluded from the first component with values below 0.5. Then, the second principal component consists of Zn and Cd; however, Zn shows a slightly peculiar pattern between HMs, as can be shown in Table 8. Finally, Figure 3a illustrates the two components as extracted by SPSS, while Figure 3b is the same picture in more focus.

## 4. Discussion

The physicochemical properties of the present four-year study have ordinary values in comparison with other studies in Greece [18,35,44–50]. However, according to Golia et al. (2021) some soil samples in the vicinity of the harbor and the promenade demonstrate higher standards of EC because of the existence of salts [18]. This means that almost 20% of the soil samples show sandy texture, probably due to the proximity of the samples to the sea (Pagasitikos Gulf), as they are mainly concentrated in the coastal zone of the city of Volos.

The total concentrations of HMs referring to the mean values and the medians during 2018–2021 follow the order: Mn > Zn > Cr > Ni > Cu > Pb > Co > Cd. Furthermore, the total concentrations of the HMs were compared with the limits of Directive 86/278/EEC (Table 3) in order to investigate potential hazardous exceedances [58]. Only Ni and Zn surpassed the higher limits of the Directive; however, they do not pose direct risks for humans in these concentrations.

Moreover, the majority of HMs could be characterized by strong and statistically significant correlation (Table 6). In addition, CA and PCA (Figures 2a and 3) exhibited the presence of two groups, the first component consisting of Mn, Pb, Cr, Cu, Ni, and Co, while

the second consists of Zn and Cd, meaning that there is potential common anthropogenic origin for the monitored HMs.

Observing Scheme 1, the samples with high concentrations of all HMs and mainly Zn, Cd and Cu are located in the area near the city and intercity bus station, the railway station and the commercial and passenger port of the city (samples: s1–s10). For example, some researchers—Christoforidis and Stamatis (2009) and Rosen (2002)—showed that concentrations of Zn and Cd could exist in car brakes, while Zn concentrations could be found in railways. Other HMs could have, as potential sources, the brakes or the fuels of car, such as Cr and Pb [18,35,40,61,62]. These HMs might aggravate the standards of urban soils, which conclusively would have negative externalities towards citizens and the environment.

Even though the average values of the HM concentrations show an upward trend, it is worth considering the change in concentration per year in each soil sample separately. Specifically, almost in the majority of the samples which are located near the main roads in Scheme 1 appear to have lower concentrations in the years 2020 and 2021 compared to the first 2 years of the study. This effect may be caused because of the movement restrictions due to the COVID-19 pandemic. Cars and vehicles in general has movement restricted along the streets as citizens were not allowed to travel for 12 h (7 p.m. to 7 a.m.).

For instance, the impact of COVID-19 on transportation might have 'translocated' the urban pollution from the boulevards and main roads to the inner residential city blocks. One might wonder why a phenomenon such as this may happen. Anastasiou and Duquenne (2021a,b) concluded that the lockdowns of COVID-19 altered people's daily routines [22–25]. In addition, Ngai et al. (2021) and Jiang et al. (2021) referred to the restrictions on traffic along the streets, as citizens were not allowed to travel, as well as the introduction of telecommuting as an alternative way of working [26,27]. Hence, according to the present study, during 2018–2021 the general pattern was a yearly positive percentage change but with a decreasing trend especially between 2019–2020 (the start of the pandemic).

Focus should be given to some HMs: for instance, Zn could be found in the proximity of trains. Specifically, during the pandemic, Zn had lower concentrations, probably because of the reduction in train routes between the prefectures due to COVID-19 restrictions. However, this flow was altered, and Zn concentrations followed an upward trend in 2021, when the restrictions were repealed.

On the other hand, in the samples located in the residential area of the city (samples: s39–s53), the reverse trend was observed. That is, in the years 2020 and 2021 there was an increase in all metals. This is probably due to the fact that, during the quarantine period, residents stayed in their homes longer, so the heating systems were operated for more hours during the day and, therefore, combustion gases were more in volume and content than compounds of metals [22–25]. These gases were naturally transported to the ground and the imprint of the pollution was recorded. It is, therefore, not unreasonable that, during the years of the pandemic, Zn and Cd concentrations in some samples were significantly altered and destabilized. This environmental destabilization could be shown in the rising trend in HM concentrations in 2020 and in 2021 for these two specific HMs during the COVID-19 era.

## 5. Conclusions

In the city of Volos, for four consecutive years, a study of soil pollution was carried out. A total of 62 soil samples were collected per year from the urban complex, from green areas along the boulevards and main roads and from areas adjacent to the bus station, trains, and the city port. Using the appropriate statistical tools (Levene's test, ANOVA, Tukey test, Fisher LSD test, Games–Howell, Kolmogorov–Smirnov, Kruskal–Wallis, CA, and PCA), a key relationship was found between the HMs of the study, which implies a possible common origin. During the four years of the study, an ever-increasing trend in the concentration of minerals was observed. However, in the years 2020 and 2021, there was a shift in pollution, with a decrease in concentrations on the wheeled vehicles and

an increase in the residential area, due to the restriction of traffic on the one hand and an increase in heating hours in citizens' homes.

Apparently, driving forces such as population growth and rampant industrialization have accelerated HM concentrations. For instance, the aforementioned drivers of change have accumulated HMs through metallurgical activities, exhausts from vehicles, and ship emissions. Accordingly, COVID-19 has abruptly changed human lives and the transportation habits due to the two lockdowns; hence, it 'transported' a part of HM concentrations from boulevards to the inner residential blocks.

Last but not least, it is advisable that further focus on urban soil pollution be given, in order to preserve and protect the natural environment and citizens' well-being. In addition, the overaccumulation of HMs over years—in air or soil—possibly robustly aggravating citizens health conditions and destabilizing physical processes, such as the provocation of soil weathering. Additionally, it is imperative that local authorities preserve urban soil conditions. In addition, it is of utmost importance that institutional factors enable their technological and economic armamentarium in order to preserve nature via regulations and action plans as well. In a nutshell, our study focuses on the depiction and analysis of urban soil pollution to sensitize every stakeholder in order to make, ultimately, a world of no waste and slight-to-minimum pollution. Further research is needed in the future, with the recording of pollution effects and the construction of thematic maps for each metal for the sake of capturing the impact of pandemic restrictions on pollution in the city center.

**Supplementary Materials:** The following supporting information can be downloaded at: https://www.mdpi.com/article/10.3390/land11071016/s1, Table S1: Output of Tukey and LSD Fisher tests for Cu.; Table S2: Output of Tukey and LSD Fisher tests for Zn.; Table S3: Output of Tukey and LSD Fisher tests for Mn.; Table S4: Output of Tukey and LSD Fisher tests for Pb.; Table S5: Output of Tukey and LSD Fisher tests for Ni.; Table S6: Output of Tukey and LSD Fisher tests for Cr.; Table S7: Output of Tukey and LSD Fisher tests for Co.; Table S8: Output of Games-Howell test for Cd.

**Author Contributions:** Conceptualization, P.-S.C.A. and E.E.G.; methodology, P.-S.C.A. and E.E.G.; software, P.-S.C.A. and E.E.G.; validation, P.-S.C.A. and E.E.G.; formal analysis, P.-S.C.A.; investigation, P.-S.C.A. and E.E.G.; resources, P.-S.C.A. and E.E.G.; data curation, P.-S.C.A. and E.E.G.; writing—original draft preparation, P.-S.C.A. and E.E.G.; writing—review and editing, P.-S.C.A. and E.E.G.; visualization, P.-S.C.A. and E.E.G.; supervision, E.E.G.; project administration; funding acquisition, P.-S.C.A. P.-S.C.A. carried out the experimental procedure as part of his post-graduate thesis, under E.E.G. supervision. All authors have read and agreed to the published version of the manuscript.

**Funding:** This research received no external funding.

**Data Availability Statement:** Data available on request due to restrictions, e.g., privacy or ethical. The data presented in this study are available on request from the corresponding author. The data are not publicly available due to privacy and copyright reasons.

**Acknowledgments:** The authors would like to express their gratitude to the Postgraduate Program of the University of Thessaly, entitled: Sustainable Management of Environmental Change and Circular Economy. The results derive from the post-graduate dissertation of student Panagiotis-Stavros C. Aslanidis. Part of the results come from the post-graduate dissertation of student Sotiria G. Papadimou, on which the former dissertation was based.

**Conflicts of Interest:** The authors declare no conflict of interest.

## Appendix A

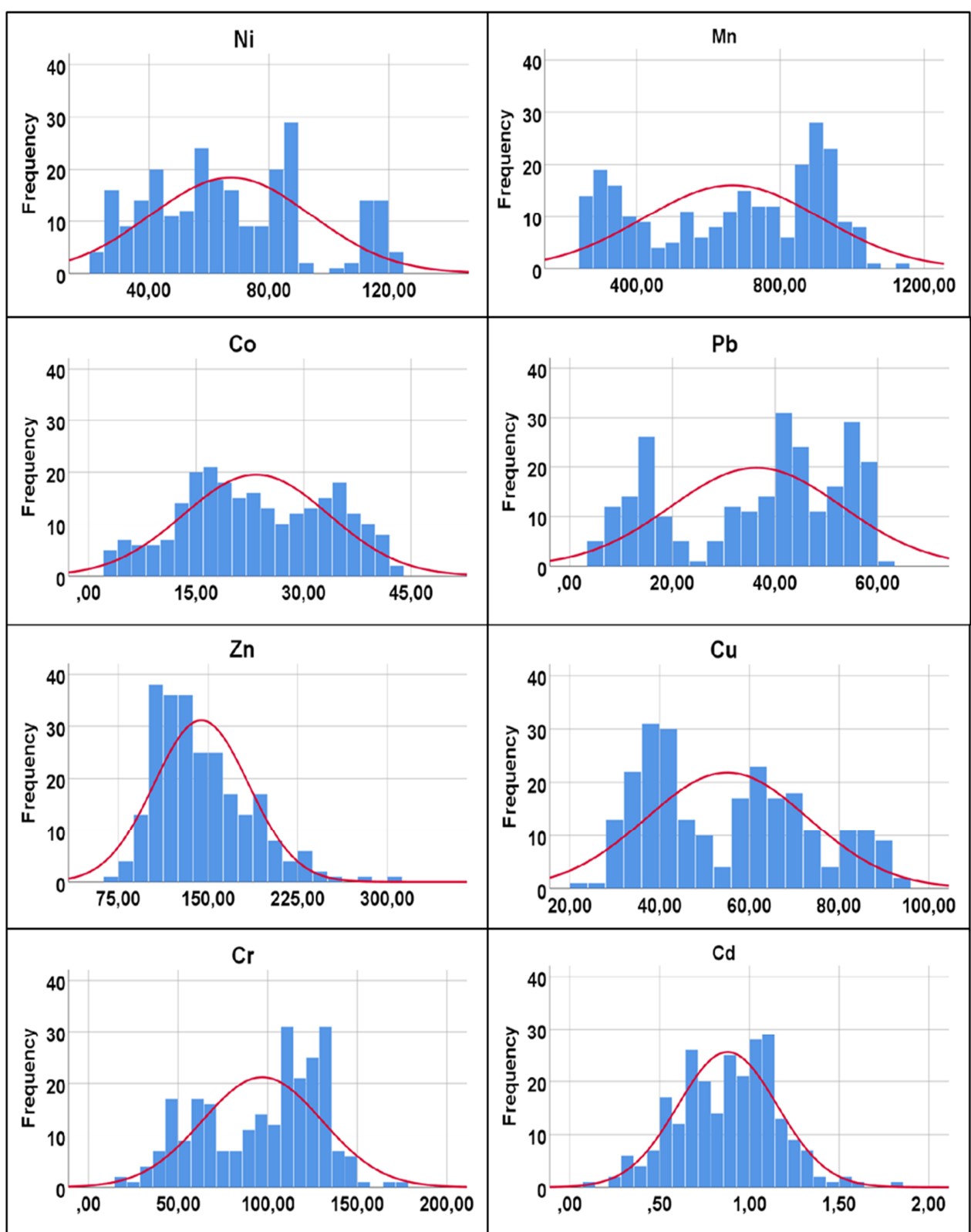

**Figure A1.** The bar charts of the eight HMs and the red lines which indicate the line of normal distribution.

**Appendix B**

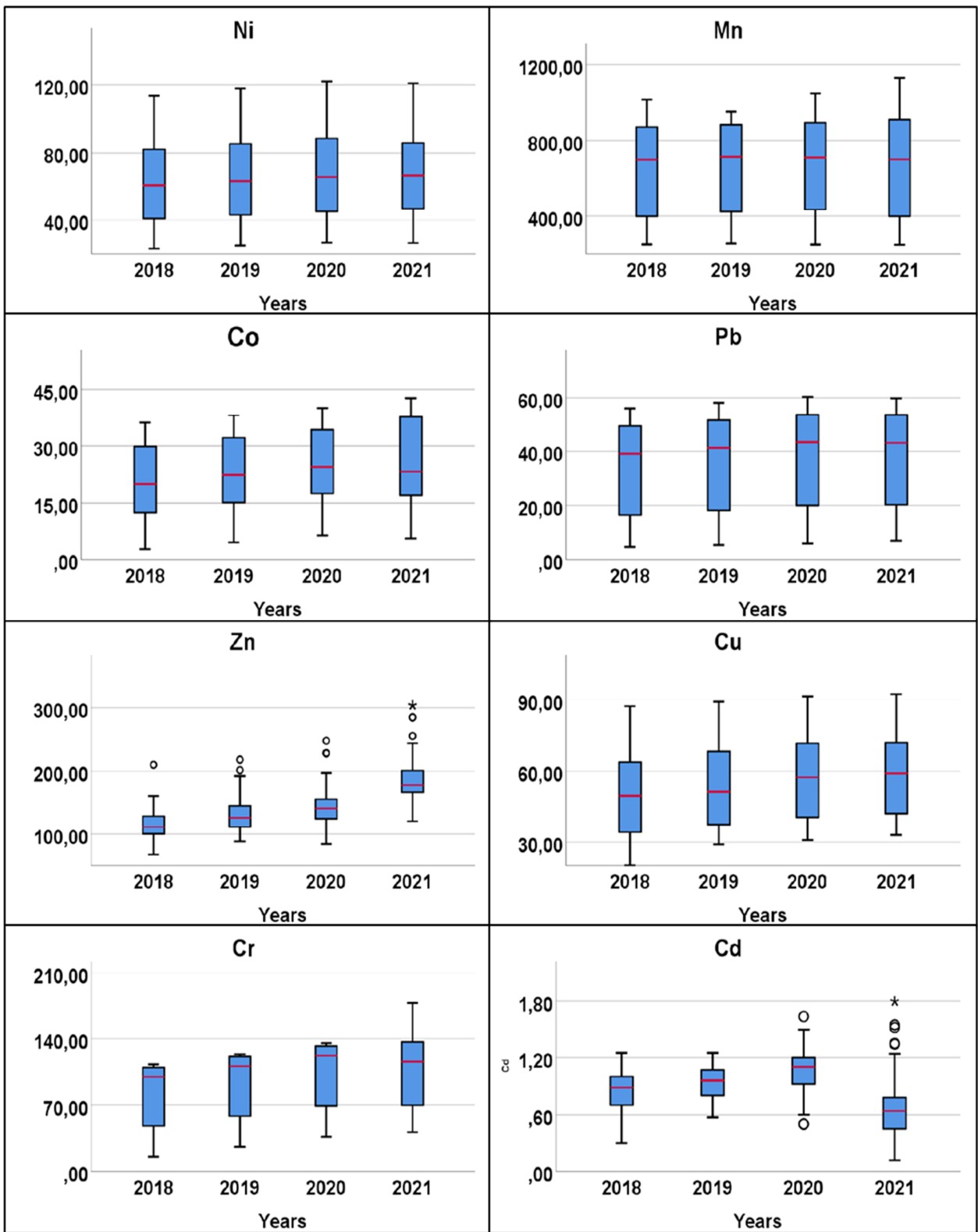

**Figure A2.** The box plots of the eight HMs and the red lines which indicate the median. The existence of circles and stars indicates that there are outliers, which are values outside the whiskers of the box plot.

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
