# Peer review of "Urban Sustainability at Risk Due to Soil Pollution by Heavy Metals—Case Study: Volos, Greece"

_land, doi:10.3390/land11071016_

Round 1
Reviewer 1 Report
Some changes were made to manuscript, which significantly improved the quality of the article. However, some corrections are advisable. In addition to minor editorial changes (punctuation errors, typos and missing spaces, bold font in some places), it is also worth explaining the GHGs abbreviation, the meaning of which I unfortunately did not find. The citation of Appendixs should be supplemented in the text. With regard to soil contamination, the methodology should provide a description of the site taking into account the geological substrate and the existing soils, as well as the lack of soil texture in the results. The results chapter does not describe the results based on groups of samples from individual areas (districts) of the city, which is the case in the discussion. It is worth introducing the reader to this description of the results, especially since the discussion includes short statements indicating the importance of the area in determining the variability of soil contamination with heavy metals. It is understandable to compare the content of heavy metals in the studied urban landscapes with the directive (which is the case in Table 3 and the relevant description), however, the pattern observed by the author is poorly described and is not well understood. On line 236 there was a mistake in the numbering of the table.
The conclusions lack a strong reference to environmental stability in relation to heavy metals, which has been highlighted in the novelty of article. This aspect should also be briefly discussed in the discussion. The last paragraph of the discussion seems to cover the literature and is worth discussing.
Author Response
Thank you for the comments, please see the attachment.

Reviewer 2 Report
This is a revised MS. It is recommended to be published at the present form.
Author Response
Thank you for the comment. Please see the attachment.

Reviewer 3 Report
the presented work deals with a topic that is very common in the literature. however, this is an important topic and needs to be addressed. Contamination of urban soils is a serious problem that threatens all urban dwellers.
The methodology of the article is very good and is clearly described. the presentation of the results is also very good. The results are well discussed. The conclusion is clear and concise.
I recommend publishing the article without editing.
Author Response

(The authors gave the same response as above.)
